# Adipose Tissue Caveolin-1 Upregulation in Obesity Involves TNF-α/NF-κB Mediated Signaling

**DOI:** 10.3390/cells12071019

**Published:** 2023-03-27

**Authors:** Ashraf Al Madhoun, Shihab Kochumon, Dania Haddad, Reeby Thomas, Rasheeba Nizam, Lavina Miranda, Sardar Sindhu, Milad S. Bitar, Rasheed Ahmad, Fahd Al-Mulla

**Affiliations:** 1Genetics and Bioinformatics, Dasman Diabetes Institute, Dasman 15462, Kuwait; dania.haddad@dasmaninstitute.org (D.H.); rasheeba.iqbal@dasmaninstitute.org (R.N.); milad.bitar@gmail.com (M.S.B.); 2Animal and Imaging Core Facilities, Dasman Diabetes Institute, Dasman 15462, Kuwait; lavina.miranda@dasmaninstitute.org (L.M.); sardar.sindhu@dasmaninstitute.org (S.S.); 3Immunology & Microbiology Department, Dasman Diabetes Institute, Dasman 15462, Kuwait; shihab.kochumon@dasmaninstitute.org (S.K.); reeby.thomas@dasmaninstitute.org (R.T.); 4Department of Pharmacology, Faculty of Medicine, Kuwait University, Jabriya 046300, Kuwait

**Keywords:** caveolin-1, obesity, adipose tissue, metabolic inflammation, cytokines, TNF-α, NF-κB

## Abstract

Obesity is characterized by chronic low-grade inflammation. Obese people have higher levels of caveolin-1 (CAV1), a structural and functional protein present in adipose tissues (ATs). We aimed to define the inflammatory mediators that influence *CAV1* gene regulation and the associated mechanisms in obesity. Using subcutaneous AT from 27 (7 lean and 20 obese) normoglycemic individuals, in vitro human adipocyte models, and in vivo mice models, we found elevated CAV1 expression in obese AT and a positive correlation between the gene expression of CAV1, tumor necrosis factor-alpha (TNF-α), and the nuclear factor kappa-light-chain-enhancer of activated B cells (NF-κB). *CAV1* gene expression was associated with proinflammatory cytokines and chemokines and their cognate receptors (*r* ≥ 0.447, *p* ≤ 0.030), but not with anti-inflammatory markers. *CAV1* expression was correlated with CD163, indicating a prospective role for CAV1 in the adipose inflammatory microenvironment. Unlike wild-type animals, mice lacking TNF-α exhibited reduced levels of CAV1 mRNA/proteins, which were elevated by administering exogenous TNF-α. Mechanistically, TNF-α induces *CAV1* gene transcription by mediating NF-κB binding to its two regulatory elements located in the CAV1 proximal regulatory region. The interplay between CAV1 and the TNF-α signaling pathway is intriguing and has potential as a target for therapeutic interventions in obesity and metabolic syndromes.

## 1. Introduction

Obesity is a complex multifactorial metabolic disorder involving genetic, socioeconomic, and environmental factors [1]. Despite significant efforts made to raise awareness, obesity remains a major global burden and is a precursor of several diseases, including type 2 diabetes mellitus (T2DM) and cancer, further contributing to a lower quality of life and higher mortality rates [2]. In Kuwait, obesity has reached endemic levels [3]. A study conducted in 2014 reported that approximately eight out of ten Kuwaiti adults were either overweight or obese, highlighting the need for urgent action plans and strategies [4].

Adipose tissue (AT) is a plastic and dynamic endocrine organ that functions as a primary fat and triacylglycerol deposit [5]. AT also possesses the plasticity to adjust to a surplus of energy through progressive adipocyte hypertrophy and proliferation. AT also does so via the recruitment and differentiation of precursor cells and mechanisms associated with active vascularization and extracellular matrix remodeling [6]. Furthermore, AT is an endocrine organ that produces and secretes hormones, adipokines, cytokines, and chemokines [5]. In obese individuals, aberrant AT and residential immune cells intensify local inflammatory responses and augment metabolic dysfunctions [7].

Caveolae are cholesterol-rich lipid rafts located in the plasma membrane as inward cytoplasmic invaginations, protruding into the cytoplasm. Although caveolae are present in many types of cells, they are particularly abundant in adipocytes, accounting for approximately 30% of the plasma membrane [8]. Caveolae are implicated in several essential cellular functions, such as endocytosis, transcytosis, maintenance of plasma membrane integrity, lipid homeostasis, signal transduction, and mechanoprotection [9,10]. Caveolin-1 (CAV1) is an integral structural and functional protein of the caveolae that helps in the recruitment of various signaling molecules and plays an indispensable role in the insulin signaling pathway [11,12]. A previous study reported an increase in CAV1 expression in the AT of obese individuals, with or without T2DM, and an association between CAV1 and several inflammatory markers, suggesting a prospective role for CAV1 in obesity-related chronic low-grade inflammation, called metabolic inflammation [13].

In obese individuals, dysfunctional adipocytes secrete proinflammatory cytokines, such as tumor necrosis factor-alpha (TNF-α) and interleukin-6 (IL-6), as well as chemokines, especially CCL2, which plays a major role in recruiting immune cells to ATs, resulting in obesity-associated metabolic inflammation [14]. This is associated with the onset of various conditions, including hyperinsulinemia, dyslipidemia, T2DM, and vascular abnormalities [15,16]. TNF-α, by activating the nuclear factor kappa-light-chain-enhancer of activated B cells (NF-κB), contributes positively toward maintaining a proinflammatory microenvironment in the adipocyte, which further results in significant changes in the expression of adipocyte-specific genes and affects their function [17,18]. Although previous studies have reported the effects of TNF-α on insulin resistance and the destabilization of insulin receptor–CAV1 interactions [19], the association between this cytokine and CAV1 expression in normoglycemic obese individuals has not been studied before.

In the context of obesity and metabolic syndromes, CAV1 regulation and association with inflammatory markers are not well defined. Therefore, in the present study, we aimed to evaluate the correlations between CAV1 and proinflammatory cytokine transcripts in non-diabetic obese individuals. Then, we thoroughly investigated the role of TNF-α-mediated CAV1 upregulation using pre-adipocytes isolated from lean and obese individuals as an in vitro model. Herein, we show, for the first time to our knowledge, that the simultaneous increase in CAV1 expression in the ATs of obese individuals correlates with inflammatory and metabolic markers in this population. We further demonstrate the reduced adipose CAV1 expression in TNF-α global knock-out (KO) mice and the positive modulatory changes in its expression by exogenous TNF-α in this mouse model. We also show that TNF-α acts by activating the NF-κB binding sites located in CAV1 upstream regulatory regions.

## 2. Materials and Methods

### 2.1. Study Population and Anthropometric Measurements

The study cohort comprised a total of 27 male and female non-diabetic adult individuals, enrolled at the gymnasium facility at the Dasman Diabetes Institute in Kuwait. Using the standard formula for calculating BMI, i.e., body weight (kg)/height^2^ (m^2^), the cohort was divided into seven lean (BMI < 25 kg/m^2^) and 20 obese (BMI ≥ 30 kg/m^2^) participants. The exclusion criteria were as follows: individuals with a history of chronic disease or medications, malignancy, and pregnancy. Anthropometric measurements, such as height, weight, and waist circumference, were measured, as described previously [20,21]. Whole-body compositions including body fat percentage, soft lean mass, and total body water were measured using an IOI 353 Body Composition Analyzer (Jawon Medical, Seoul, Republic of Korea). The biochemical and demographic characteristics of the participants are summarized in Table 1. This study was performed in accordance with the Declaration of Helsinki and was approved by the Ethics Committee of the Dasman Diabetes Institute, Kuwait. Written informed consent was obtained from each participant prior to the study.

### 2.2. Collection of Subcutaneous AT

We collected AT samples (approximately 500 mg) via abdominal subcutaneous fat pad biopsy by following standard surgical procedures, as described previously [20,22,23]. Briefly, the periumbilical area was decontaminated using alcohol-soaked gauze, and then locally anesthetized using 2% lidocaine. Fat tissue was collected through a small superficial skin incision (0.5 cm). Next, the biopsy tissue was further incised into smaller sections, washed with cold phosphate-buffered saline (PBS), preserved in RNAlater solution, and stored at −80 °C until use [24].

### 2.3. Blood Collection and Biochemical Measurements

Peripheral blood samples (~10 mL each) were obtained from the participants after an overnight fast; the samples were collected in vacutainer EDTA tubes, and plasma was collected by centrifugation, aliquoted, and stored at −80 °C until assayed [25]. Fasting blood glucose (FBG) levels and lipid profile, including total cholesterol, low-density lipoprotein (LDL), high-density lipoprotein (HDL), and triglycerides, were measured using the Siemens Dimension RXL Chemistry analyzer (Diamond Diagnostics, Holliston, MA, USA). The Homeostatic Model Assessment of Insulin Resistance (HOMA-IR) was calculated from basal (fasting) glucose and insulin concentrations using the following formula: HOMA-IR = fasting insulin (μU/L) × fasting glucose (nmol/L)/22.5. HbA1c was measured using the Variant device (BioRad, Hercules, CA, USA). White blood cells were counted using a hematocytometer. All assays were performed as per the instructions provided by the manufacturers.

### 2.4. RNA Extraction, cDNA Synthesis, and Reverse Transcription-Quantitative Polymerase Chain Reaction (RT-qPCR)

Total RNA was extracted from AT using the RNeasy kit (Qiagen, Valencia, CA, USA), following the manufacturer’s protocol. The first cDNA strand was synthesized from 0.5 µg RNA using the High-Capacity cDNA Reverse Transcription kit (Applied Biosystems, Waltham, MA, USA). Real-time RT-qPCR was performed as described previously [26]. cDNA samples (50 ng each) were amplified using TaqMan Gene Expression Master Mix (Applied Biosystems) and gene-specific 20× TaqMan gene expression assays (Applied Biosystems), containing forward and reverse primers (Appendix A), along with the target-specific TaqMan MGB probes labeled with FAM dye at the 5′-end and NFQ-MGB at the 3′-end of the probe using a 7500 Fast Real-Time PCR System (Applied Biosystems). Each cycle involved denaturation for 15 s at 95 °C; a uracil-DNA glycosylases cycle for 2 min at 50 °C; an annealing/extension for 1 min at 60 °C, and AmpliTaq gold enzyme activation for 10 min at 95 °C. Gene expression relative to lean AT (control) was calculated using the comparative cycles to the threshold (C_T_) method, as previously described [27]. The results were normalized to glyceraldehyde 3-phosphate dehydrogenase (GAPDH) expression, and data (mean ± SEM) were expressed as fold changes in expression relative to controls, as indicated [28].

### 2.5. Cell Culture, DNA Transfections, and Luciferase Reporter Assays

Pre-adipocytes isolated from lean and obese individuals were purchased from Zen-Bio (Durham, NC, USA). Cells were grown in DMEM/F12 media (Invitrogen Corporation, Carlsbad, CA, USA) supplemented with 10% fetal bovine serum (Invitrogen Corporation) and 100 U/mL of penicillin–streptomycin (Gibco, Carlsbad, CA, USA), and were incubated at 37 °C in 5% CO_2_. For luciferase reporter assays, cells were plated at a density of 5000 cells/well in 96-well plates and were transiently transfected using lipofectamine LTX (Thermo Fisher, Waltham, MA, USA) with pRMT-Luc-CAV1, which contains CAV1 proximal promoter, or pRMT-Luc vector. Transfection efficiency was monitored by analyzing Renilla–luciferase activity, as described previously [29]. After 24 h, the cells were treated with TNF-α (10 ng/mL; Sigma, Saint Louis, MO, USA), or vehicle. Twenty-four hours post-treatment, the cells were washed twice with ice-cold PBS, lysed, and assayed in accordance with the Dual Luciferase Kit protocol (Promega, Madison, WI, USA). Luciferase activity was detected using an assayed GloMax Navigator Microplate Luminometer (Promega). To remove the background, the normalized activity of the luciferase reporter vector alone was subtracted from its activity in the presence of TNF-α. Relative luciferase units (RLUs) represent luciferase activity normalized to Renilla activity [25,30].

### 2.6. Chromatin Immunoprecipitation (ChIP) Assays

Chromatin immunoprecipitation (ChIP) assays were performed, as described previously [31], with minor modifications. Pre-adipocytes from lean and obese individuals were seeded on three 175 mm tissue culture flasks with 1 × 10^6^ cells/flask. After 48 h, 50 mg of chromatin was immunoprecipitated in each cell line using 0.5 µg of anti-NF-κB-specific antibodies (L8F6, Cell Signaling Technology, Danvers, MA, USA), anti-Histone-3 acetyl K14 antibodies (H3K14^ac^, ab176799, Abcam, Cambridge, MA, USA), or non-specific rabbit IgG control antibodies (ab172730, Abcam). Briefly, the cells were crosslinked with 4% formaldehyde (Fisher Scientific, Hampton, NH, USA) and sonicated using a Covaris-E220 ultrasonicator (Covaris, Wohurn, MA, USA) for a total of 30 × 15 s pulses (1 min rest between the pulses), and lysates were cleared by centrifugation at 13,000 rpm for 30 min at 4 °C. The sheared chromatins were incubated with the described antibodies, and the immune complexes were captured using protein G-Sepharose Dynabeads (ThermoFisher, Waltham, MA, USA), as described previously [32]. NF-κB, H3K14^ac^, or IgG-bound chromatins were quantified as a percent of chromatin input. Quantitative PCR (qPCR) analyses were performed using the forward 5′-GAGGTGAAGAGAAGCCAGGAAT-3′ and reverse 5′-CCCAATCTCAGGACCCCAAT-3′ primers. To confirm a true association, each ChIP sample was examined for the enrichment of a chromatin locus immunoprecipitated with a specific antibody and compared with the same chromatin locus immunoprecipitated with a non-specific IgG. Data are represented as mean ± SEM from three independent biological experiments [33].

### 2.7. Immunohistochemistry (IHC) Assays

IHC staining was performed, as described previously [34,35]. Paraffin-embedded sections (4 μm thick) of subcutaneous AT were deparaffinized in xylene and rehydrated through descending grades of ethanol (100%, 95%, and 75%) to water. Antigen retrieval was then performed by placing slides in a target retrieval solution (pH 6.0; Dako, Glostrup, Denmark) in the pressure cooker, boiling for 8 min, and cooling for 15 min. After washing in PBS, endogenous peroxidase activity was blocked with 3% H_2_O_2_ for 30 min and non-specific antibody binding was blocked with 5% non-fat milk for 1 h, followed by 1% bovine serum albumin solution for 1 h. The slides were incubated at room temperature overnight with primary antibodies, such as rabbit polyclonal anti-CAV1 antibody (3238s, Cell Signaling Technology, Danvers, MA, USA) and TNF-α antibody (ab9635, Abcam). After washing with PBS (0.5% Tween), slides were incubated for 1 h with goat anti-rabbit conjugated with horseradish peroxidase polymer chain DAKO EnVision Kit (Dako, Glostrup, Denmark), and color was developed using a 3,3′-diaminobenzidine (DAB) chromogen substrate. The specimens were washed, counterstained, dehydrated, cleared, and mounted, as described elsewhere [36]. For analysis, digital photomicrographs of the entire AT sections (20×; Pannoramic Scan, 3DHistech, Budapest, Hungary) were used to quantify the immunohistochemical staining using ImageJ software (NIH, Bethesda, MD, USA). CAV1 and TNF-α antibody specificity was validated using spleen tissue, as shown in Appendix A.

### 2.8. Western Blotting and Adipogenesis Protocols

Western blot analysis was performed, as described previously [26]. Briefly, primary pre-adipocytes were differentiated into mature adipocytes using the differentiation media supplied by Zen-Bio as described by the manufacturers. Briefly, 70% confluent pre-adipocytes were incubated in Adipocyte Differentiation Medium (DM-2) for 7 days at 37 °C, and 5% CO_2_. On day 7, the DM-2 was replaced with Adipocyte Medium (AM-1) and incubation was continued for another 5–7 days. DM-2 and AM-1 were frequently changed every 2 days. Adipocytes were monitored for lipid droplet appearance in the cytoplasm as described in the instruction manual number ZBM0001.05 from Zen-Bio (Appendix A). The differentiation authenticity was detected by Western blot assays using antibodies against the lipid droplet marker perilipin-1 (9349s, Cell Signaling Technology, Danvers, MA, USA) and the adipogenesis marker adiponectin (2789s, Cell Signaling Technology, Danvers, MA, USA), as shown in Appendix A.

Adipocytes were treated with TNF-α (10 ng/mL, Sigma) or vehicle. After 24 h of treatment, cells were harvested for total RNA and protein preparations, as described previously [27]. For protein extracts, cells were harvested and lysed in RIPA buffer. Cell lysates were quantified using the Quick start Bradford assay (Bio-Rad, Hercules, CA, USA), and equal amounts of protein were resolved on 8–12% polyacrylamide gels and transferred to Nitrocellulose membranes (Bio-Rad, Hercules, CA, USA). After blocking, membranes were blotted with the following primary antibodies: CAV1, 3238s; β-actin, 4970L; TNF-R1 (C25C1), 3736s; IkB-alpha (L35A5), 4814s; NF-κB P65 (D14E12), 8242s; Phospho-NF-κB p65 (Ser536) (93H1), 3033s; (Cell Signaling Technology, Danvers, MA, USA) and the corresponding horseradish peroxidase-linked secondary antibody (cat. #7074P2, Cell Signaling Technology). Proteins were visualized using a SuperSignal West Femto ECL kit (Thermo Scientific, Waltham, MA, USA). Images were captured using the ChemiDoc MP imaging system (Bio-Rad, Hercules, CA, USA).

### 2.9. Mice

Male mice (6–7 weeks old; 23.64 ± 2.76 g) were housed at the Animal Core Facility, Dasman Diabetes Institute, Kuwait. The mice were housed in temperature-controlled rooms (22 °C) with a 12 h light/dark cycle and access to standard laboratory food and water ad libitum. All experiments on animals were approved by the ethics committee for the use of Laboratory Animals in Teaching and in Research, Dasman Diabetes Institute, Kuwait, in accordance with the guidelines of the Animal Research: Reporting of in vivo Experiments (ARRIVE). Wildtype control mice (B6129SF2/J, Strain #:101045, RRID: IMSR_JAX:101045) and TNF-α^−/−^ mice (B6;129S-Tnftm1Gkl/J, Strain #:003008, RRID: IMSR_JAX:003008) were acquired from the Jackson Laboratory (Bar Harbor, ME, USA). Both genotypes were injected intraperitoneally with either physiological saline or 50 µg/kg TNF-α for five days, followed by 100 µg/kg TNF-α for three days. After a total of eight treatment days, mice were scarified. To delineate their CAV1 mRNA and protein expression levels, subcutaneous AT samples were harvested, as previously described [7].

### 2.10. Live-Staining of Lipid Droplets with NILE Red

Nile red powder (N1142, Invitrogen, Carlsbad, CA, USA) was dissolved in DMSO for a stock concentration of 1 mM, according to manufacturer protocol. Differentiated cells were washed with PBS, covered with 1/100 dilution of Nile red, and incubated for 15 min in the dark. Cells were washed 3 times with PBS. Images were acquired with a Zeiss Stereo microscope (Discovery V12, Jana, Germany) at 552/636 nm.

### 2.11. Statistical Analyses

Statistical analyses were performed using GraphPad Prism software (La Jolla, San Diego, CA, USA) and SPSS for Windows version 19.01 (IBM SPSS Inc., Armonk, NY, USA) as described previously [37]. Data are shown as mean ± standard deviation, unless otherwise indicated. An unpaired Student’s *t*-test was used to compare the means between the treatments and prospective controls. Pearson correlation was used to determine associations between different variables. For all analyses, a *p*-value < 0.05 was considered significant.

## 3. Results

### 3.1. Demographic and Clinical Characteristics of the Study Population

The demographic and clinical characteristics of the 27 participants included in this study are shown in Table 1. The mean age of the study participants was 44 years; weight, BMI, and body fat percentage were significantly higher in the obese participants compared to the lean individuals (*p* < 0.0001 for all). The mean values of total plasma cholesterol and LDL levels were comparable between the two groups, whereas, in obese individuals, plasma triglycerides and HDL levels were marginally higher and lower, respectively (*p* = 0.069 and *p* = 0.071, respectively). All participants had a normal or prediabetic status, with FBG and HbA1c measurements at 5.24 ± 0.53 mmol/L and 5.7 ± 0.50%, respectively, for lean individuals, and 5.20 ± 0.63 mmol/L and 5.59 ± 0.67%, respectively, for obese subjects. Expectedly, HOMA-IR values were also elevated, averaging between 1.51 ± 0.69 and 2.59 ± 1.86 for lean and obese participants, respectively. The plasma insulin concentrations were below 20 mU/L, i.e., within normal fasting concentrations, in both groups.

### 3.2. Elevated CAV1 Gene and Protein Expression in Subcutaneous AT from Obese Individuals

RT-qPCR analysis showed a significant increase (1.7-fold) in *CAV1* mRNA expression in subcutaneous ATs isolated from obese subjects as compared to lean individuals (Figure 1A). Similarly, IHC analysis of CAV1 protein expression in AT sections showed higher protein levels in obese as compared to lean participants (Figure 1B). The quantification of CAV1 protein expression in AT showed a two-fold increase in CAV1 expression during IHC analysis of FFPE-stained AT from obese individuals as compared to lean individuals (Figure 1C).

### 3.3. Correlations between CAV1 Expression and Inflammatory Markers in Obese AT

To evaluate the role of CAV1 in obesity-associated metabolic inflammation, we analyzed the association between *CAV1* gene expression and the expression of various inflammatory cytokines in obese participants. As shown in Table 2, RT-qPCR analyses revealed a notable positive correlation between *CAV1* gene expression levels and those of several C-C motif chemokines, such as *CCL2* and *CCL8* and their receptors *CCR1* and *CCR2* (0.491 ≤ r ≤ 0.618; *p* < 0.035; Table 2), which are known to be expressed in AT-residential macrophages and lymphocytes and are implicated in obesity-induced low-grade chronic inflammation [38]. Moreover, a significant correlation was observed between the transcript levels of *CAV1* and members of the angiostatic C-X-C motif chemokines—*CXCL9*, *CXCL10*, and *CXCL11* (*p* = 0.016, *p* = 0.029, and *p* = 0.017, respectively). When studying the correlation between *CAV1* and interleukins, a significant correlation was observed between *CAV1* expression and that of interleukin 1 receptor type 1 (*IL1RL1*) (*p* = 0.009); no significant correlation between *CAV1* gene expression and any other interleukin was detected (Table 2). Remarkably, *CAV1* transcript levels were not associated with that of the β-cell maturation and differentiation markers *IL-5* or *IL-13*, the Th1 cell marker *IL-12A*, or the anti-inflammatory *IL-10* (Table 2). Together, these results indicate that CAV1 is only associated with pro-inflammatory and not anti-inflammatory markers. When evaluating the correlation between *CAV1* and M1/M2 macrophage markers, we found a significant correlation between *CAV1* and *CD163*; no such relationship was found with other macrophage subtypes. Moreover, RT-qPCR analyses show a positive correlation between *CAV1* and the toll-like receptors (TLRs) signaling cascade, such as *TLR3* (*r* = 0.509; *p* = 0.031) and *TLR4* (*r* = 0.488; *p* = 0.058); however, no such correlation was found with the anti-inflammatory *TLR9*, which is highly expressed in AT (Table 2) [39,40,41]. Additionally, we were not able to detect any correlation between *CAV1* expression and that of the signal transduction mediator downstream of the TLR signaling pathway (Table 2).

### 3.4. Correlations between CAV1 Gene Expression and TNF-α Signaling Pathway

A significant correlation was observed between the proinflammatory cytokine *TNF-α* and *CAV1* at the transcriptional level in the AT obtained from obese individuals (*r* = 0.547; *p* < 0.025; Table 2, Figure 2A). In addition, multiple stepwise regression analysis showed that *TNF-α* is independently and significantly associated with *CAV1* expression (*β* = 0.06; *p* = 0.032). At a similar trend, the transcripts of the TNF-α downstream effector *NF-κB* significantly correlated with those of *CAV1* (*r* = 0.498; *p* = 0.033; Table 2 and Figure 2B).

Next, we also performed IHC analysis using TNF-α-specific antibodies to further study the notable association at the protein level. Firstly, we observed a 3.5-fold increase in TNF-α protein levels in samples obtained from obese individuals, as compared to those from lean participants (*p* = 0.0001; Figure 2C,D). Secondly, we found that protein expression of CAV1 and TNF-α were significantly correlated (*p* < 0.0001; Figure 2E).

### 3.5. Adipose CAV1 mRNA Levels in Murine Models

Data obtained from humans indicated that the TNF-α pathway is significantly associated with CAV1 expression with perspective implication in its regulation; therefore, we aimed to determine the expression levels of CAV1 in the subcutaneous ATs obtained from wild-type and TNF-α KO mice that were intraperitoneally injected with either saline or TNF-α.

In TNF-α KO mice, there is a notable reduction in the CAV1 transcripts and proteins level, but exogenous administration of TNF-α leads to a significant increase in CAV1 expression (Figure 3). In contrast, wild-type mice showed only a moderate increase in CAV1 molecules following TNF-α administration, suggesting that TNF-α may have a more significant effect on CAV1 expression in the absence of endogenous TNF-α production (Figure 3). Overall, these findings suggest a potential mechanism for how TNF-α may regulate CAV1 expression in vivo and could have implications for understanding the role of TNF-α in various physiological processes mediated by CAV1.

### 3.6. In Vitro Analysis of CAV1 mRNA and Protein Expression in TNF-α-Treated Differentiated Adipocytes

In vivo correlation studies indicate an interplay between CAV1 and pro-inflammatory factors, particularly TNF-α. Therefore, we next performed the in vitro analyses to specifically determine the impact of TNF-α on CAV1 expression and delineate the regulatory mechanism involved. Toward this end, differentiated adipocytes obtained from lean and obese individuals were treated with vehicle or TNF-α. Notably, in vehicle-treated cells, CAV1 mRNA and protein levels were significantly elevated in obese adipocytes as compared to lean adipocytes (Figure 4). Secondly, treatment with TNF-α significantly augmented the expression of CAV1 in adipocytes, as compared to the control (Figure 4A–C). It is noteworthy that the expression levels of CAV1 mRNA and protein were comparable post-treatment with TNF-α.

Moreover, treatment with TNF-α activated the NF-κB signaling pathway in both lean and obese adipocytes. This is indicated by the increase in NF-κB phosphorylation levels (Figure 4D). Simultaneously, the expression of IκBα, which is an inhibitor of NF-κB, was notably reduced after TNF-α treatment. The reduction in IκBα expression may have contributed to the activation of NF-κB. Furthermore, TNFR1 expression was also reduced after TNF-α treatment, which suggests that cells may become desensitized to TNF-α signaling post 24 h treatments. Overall, these results suggest that TNF-α treatment modulates CAV1 expression in human adipocytes through the NF-κB pathway.

### 3.7. ChIP Analyses and Luciferase Reporter Assays

Since CAV1 gene expression is correlated with that of TNF-α and NF-κB, and TNF-α induces CAV1 transcripts and proteins in both adipocytes (Figure 4A,B) and pre-adipocytes (Figure 5A,B), we conducted an in silico analysis of the CAV1 proximal regulatory region. The analysis revealed two adjacent NF-κB binding sites spanning the 1068 bp to 1094 bp upstream of the translation state side, suggesting that TNF-α signaling could target CAV1 (Figure 5C). Hence, we performed ChIP experiments to assess NF-κB promoter binding using pre-adipocytes from lean and obese individuals.

To evaluate the role of TNF-α in mediating NF-kB binding to the regulatory region of the CAV1 gene in lean and obese pre-adipocytes, we treated lean pre-adipocytes with and without TNF-α and performed ChIP-qPCR using an NF-κB antibody. We observed a significant enrichment of chromatin fragments in TNF-α-treated lean cells (9.1-fold), relative to non-treated lean pre-adipocytes (Figure 5D). This suggests that TNF-α mediates NF-κB binding to its consensus sites located in the 5′ regulatory region of the *CAV1* gene.

In comparison, obese pre-adipocytes with no TNF-α treatment showed a much higher enrichment of chromatin fragments (22.3-fold) as compared to lean control samples (Figure 5D), indicating that elevated levels of TNF-α in the obese cells are sufficient to enhance NF-κB binding to the CAV1 regulatory region. Overall, the study suggests that TNF-α plays a role in mediating NF-κB binding to its consensus sites located at the 5′-regulatory regions of the CAV1 gene, and this effect is more pronounced in obese pre-adipocytes.

To determine if histone acetylation levels were changed, ChIP was performed with antibodies against acetylated H3K14 (H3K14^ac^), indicative of actively transcribed chromatin. ChIP-qPCR analysis revealed significantly higher chromatin fragments at the *CAV1* regulatory region, harboring the NF-κB binding side, in TNF-α-treated lean (10-fold) relative to non-treated cells (Figure 5E). Likewise, obese pre-adipocytes showed a 13-fold chromatin enrichment compared to untreated lean cells (Figure 5E). These findings suggest that TNF-α facilitates histone acetylation of the CAV1 proximal promoter, potentially contributing to increased transcription of the CAV1 gene in both lean and obese pre-adipocytes. Overall, this study provides insight into the molecular mechanisms underlying the regulation of the CAV1 gene by TNF-α and highlights the importance of histone modifications in gene expression control.

To further confirm the ChIP-qPCR data and to determine if TNF-α acts through NF-κB to enhance promoter activity of the CAV1 gene, we performed a luciferase reporter assay. First, we cloned 250 bp of the CAV1 regulatory region flanking the two NF-κB binding sites into a luciferase reporter construct and transfected it into lean and obese pre-adipocytes (Figure 5F). The cells were then either treated with a vehicle or TNF-α.

TNF-α significantly increased luciferase reporter activity in both lean and obese pre-adipocytes compared to untreated controls, indicating that the TNF-α pathway induces CAV1 gene expression (Figure 5G).

Taken together, these findings support the hypothesis that TNF-α mediates NF-κB binding to the CAV1 gene regulatory region, resulting in increased transcriptional activity and higher expression levels of CAV1 in pre-adipocytes.

## 4. Discussion

In the present study, we aimed to investigate the association between *CAV1* and proinflammatory cytokines in the AT from normoglycemic lean and obese individuals, to determine the independent predictors that may influence *CAV1* gene regulation, and to further elucidate the role of CAV1 in obesity and the regulatory mechanism involved. We identified a crosstalk between *CAV1* and a wide range of proinflammatory cytokines and chemokines, particularly, the TNF-α signaling cascade. Human in vivo analyses of AT showed that the mRNA and protein expression of CAV1 were positively correlated with that of TNF-α. Furthermore, in vitro analysis revealed that CAV1 is notably upregulated by the TNF-α signaling cascade through the binding of NF-κB to the *CAV1* promoter, as illustrated in Figure 6. In accordance with other studies, our data showed that treatment with TNF-α induced the NF-κB signaling pathway [42,43,44], augmented NF-κB phosphorylation, and attenuated IκBα expression. Notably, the expression of TNFR1 expression is reduced, suggesting a potential desensitizing process due to the prolonged TNF-α treatment, as shown in a previous study [45]. Moreover, intraperitoneal administration of TNF-α augmented the CAV1 expression levels in murine ATs. This suggests a strong association between CAV1 and different proinflammatory cytokines in obesity-mediated inflammation, particularly with TNF-α as an independent predictor.

Several studies have provided strong evidence demonstrating the role of CAV1 in the pathophysiology of various diseases, particularly regarding the increase of CAV1 expression in obesity. However, the exact mechanism by which CAV1 expression is induced remains to be elucidated [11,46]. Briand et al. reported that *CAV1* expression is crucial to increase caveolae density, accommodate larger lipid droplets, and improve glucose utilization to promote cell expansion in both in vitro and mice models. In humans, they found an initial increase in *CAV1* expression in response to overfeeding [47]. Our results are in consensus with the above findings; in the present study, the mRNA and protein expression of CAV1 was significantly augmented in obese participants and, although not significant, increased plasma triglyceride levels were seen in these subjects. Thus, CAV1-enriched adipocytes in obese individuals might have a greater capacity for lipid storage, which is a characteristic clinical trait of obesity. This notion was previously supported by Catalán et al. who found significantly higher *CAV1* expression in visceral and subcutaneous AT of obese individuals [13].

Previous studies, as well as our findings, show that TNF-α augments CAV1 expression in human obese AT. However, we demonstrated the underlying mechanism by showing the direct binding of NF-κB to the *CAV1* proximal promoter, a process that is enhanced by the TNF-α signaling cascade. In addition, CAV1 transcript and protein levels were notably reduced in TNF-α KO mice that were rescued post-exogenous TNF-α administration. In obesogenic states, TNF-α is synthesized by both adipocytes and infiltrated macrophages, acting as an endocrine and paracrine mediator through interaction with type I and II TNF-α receptors and activating NF-κB-mediated signaling [48,49]. The crosstalk between CAV1 and TNF-α in human and animal models has been previously reported. In human airway smooth muscles, Sathish et al. showed that *CAV1* is associated with inflammation and its mRNA expression is induced by TNF-α and IL-1β, which was attributed to MAP kinases and NF-κB signaling [50]. Alternatively, in patients with T2DM and diabetic neuropathy, Zhu et al. reported a negative correlation between monocytes-CAV1 and TNF-α plasma levels [51]. In the rat pancreatic β-cell model, INS-1, and CAV1 protein levels were significantly elevated post-co-treatment with TNF-α and IL-1β [52]. Palacios-Ortega et al. [L4] found that although TNF-α limited the degree of differentiation of the mouse 3T3-L1 pre-adipocytes and decreased the protein expression of CAV1, IR, and GLUT-4 during differentiation, it did not significantly reduce the mRNA expression of these genes, indicating a merely delayed state of adipocytic differentiation, as opposed to complete blockage in their activity. Moreover, TNF-α showed a continued increase in CAV1 expression in late-mature adipocytes. Similarly, in the present study, in lean differentiated pre-adipocytes induced with TNF-α, the levels of *CAV1* mRNA expression were higher than the protein expression, suggesting prospective post-transcriptional and/or post-translational regulation mechanisms (Figure 4).

Studies have found that macrophage infiltration occurs at a later stage of AT expansion and contributes toward sustained states of chronic inflammation [53], characterized by elevated expression of chemokines that are predominantly regulated by the NF-κB stimulatory signaling [54]. In our study, we found a positive correlation between *CAV1* expression and CXCLs (*CXCL9*, *CXCL10*, and *CXCL11*); this could be explained by the proinflammatory role and heightened presence of CXCLs in obesity [55]. TLRs are immune receptors well known for their role in mediating inflammation and triggering obesity and other metabolic syndromes; the activation of TLRs results in a signal transduction cascade leading to the activation of NF-κB [39]. Our team has already shown that the expression of several TLRs (*TLR2*, *TLR4*, *TLR8*, and *TLR10*) and their downstream signaling molecules are augmented in obesity, with or without T2DM, and that these higher expression levels of inflammatory mediators are directly associated with insulin resistance, BMI, M1 macrophage polarization markers, and chemokine/cytokine expression [36,56,57,58]. In the present study, we found that *CAV1* had a positive correlation with *TLR3* and *TLR4*, both of which are known for inducing proinflammatory responses. CAV1 adipose expression was also positively associated with TLRs’ downstream mediator NF-κB, along with IL1R1, further strengthening the inkling that CAV1 has associations specifically with proinflammatory cytokines and related pathways. AT macrophages contribute towards the inflammatory microenvironment of adipocytes and are multifaceted as they possess both pro- (M1) and anti-inflammatory (M2) properties. The ability of macrophages to switch their polarized state, i.e., from M2 to M1, under the activation of chemotactic molecules has been reported in murine models. In the present study, we show that *CD163*, a scavenger receptor that is highly expressed in macrophages and shed by inflammatory stimuli, was also positively correlated with *CAV1* transcripts. Obesity has previously been linked to the accumulation of CD11c^+^ CD163^+^ macrophages in both visceral and subcutaneous AT [59]. The hypertrophied state of adipocytes in our obese participants may explain these elevated levels of *CD163* expression.

Considering these intricately interwoven activities occurring in both T2DM and obesity, we infer that CAV1 may be the central factor directly or indirectly exerting a modulating effect on the increased serum triglyceride and free fatty acid levels, hyperinsulinemia, impaired glycogen storage, and a proinflammatory microenvironment in adipocytes. Thus, we provide a rationale behind its abundance in adipocytes and positive correlations with pro-inflammatory factors in adipocytes of obese normoglycemic subjects included in our study. Although the present study provides important insights, there are a few limitations. Because obesity is a multifactorial condition that is regulated by various factors, the sample size of the study is too small to generalize these findings, considering the differences in diet, lifestyle, and genetics across geographical locations. In the future, multicenter studies with participants from various backgrounds receiving different diets would be helpful to validate our findings.

## 5. Conclusions

In conclusion, we found augmented CAV1 mRNA and protein levels in the ATs of obese human participants and a positive correlation between CAV1 and proinflammatory cytokines. The TNF-α/NF-κB signaling cascade is the key mediator for CAV1 upregulation in adipocytes. Over the past decades, the CAV1 protein, owing to its role in regulating various signaling cascades, has shown immense potential as a therapeutic target for obesity and obesity-associated metabolic inflammation. Future studies with a larger sample size are warranted to decipher the exact crosstalk between CAV1 and TNF-α and the mechanism by which CAV1 plays an important role in transmitting signals from the cell surface via intracellular signaling pathways that regulate inflammation in obesity. Targeting these pathways in humans may lead to a reduction in inflammatory states, thereby reducing insulin resistance and comorbidities and improving the quality of life of patients with obesity.

## Figures and Tables

**Figure 1 cells-12-01019-f001:**
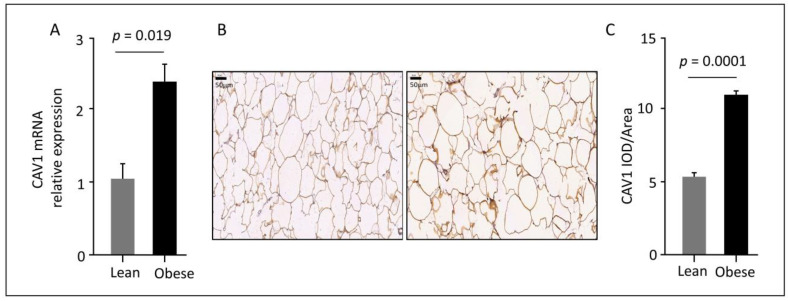
Increased CAV1 mRNA and protein levels in AT. (**A**). *CAV1* mRNA expression in AT isolated from lean and obese individuals, as determined by RT-qPCR assay (**B**). Representative images of IHC staining for CAV1 protein in tissue sections of AT obtained from lean and obese individuals. Obese sections show stronger staining than lean sections (images, magnification, ×20). (**C**). IOD/area quantification of CAV1 expression in AT from lean and obese individuals. CAV1 protein expression in obese individuals is two-fold higher, as compared to lean participants. Abbreviations: AT, adipose tissue; CAV1, caveolin-1; IHC, immunohistochemistry; IOD, integrated optical density; RT-qPCR, reverse transcription-quantitative polymerase chain reaction.

**Figure 2 cells-12-01019-f002:**
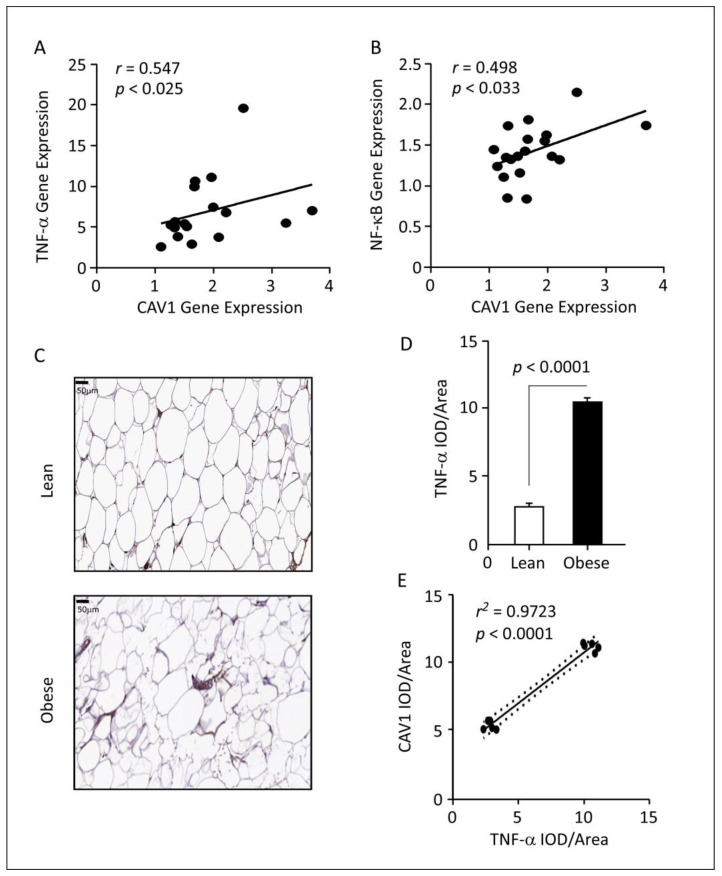
Gene expression and immunohistochemical results in AT isolated from lean and obese individuals. Correlation between *CAV1* and (**A**). *TNF-α* and (**B**)*. NF-κB* transcripts expression. (**C**). Representative images of IHC staining for TNF-α protein in tissue sections of AT obtained from lean and obese individuals. Obese sections show stronger staining than lean sections (images, magnification, ×20. (**D**). IOD/area quantification of TNF-α expression in AT from lean and obese individuals. TNF-α protein expression in obese individuals is three-fold higher compared to lean participants. (**E**). CAV1 and TNF-α immunohistochemistry IOD/area correlation. Immunohistochemical analysis was performed on samples from five individuals, and each point is the average of five different images. Abbreviations: AT: adipose tissue; CAV1: caveolin-1; TNF-α: tumor necrosis factor-alpha; NF-κB: nuclear factor kappa-light-chain-enhancer of activated B cells; IHC, immunohistochemistry, IOD: integrated optical density.

**Figure 3 cells-12-01019-f003:**
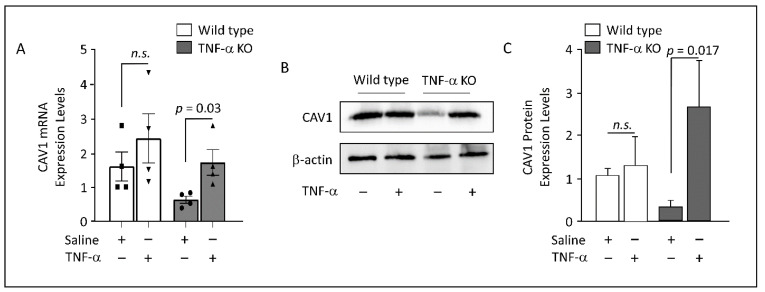
Exogenous administration of TNF-α induces CAV1 expression in vivo. (**A**). The levels of mRNA of *CAV1* in subcutaneous adipose tissue obtained from wild-type and TNF-α knock-out mice injected intraperitoneally with saline or TNF-α solution (100 µg/kg). Data are presented as mean ± SEM values of four animals/group. Two-tailed unpaired Student’s *t*-test was used to determine significance. (**B**,**C**). Western blots for total protein extracts (10 µg) were developed with antibodies against CAV1 and β-actin, as indicated. CAV1 expression was significantly elevated in response to TNF-α treatments. Two-tailed unpaired Student’s *t*-test was used to determine significance. Statistical analyses were relative to untreated wild-type animals (*n* = 4).

**Figure 4 cells-12-01019-f004:**
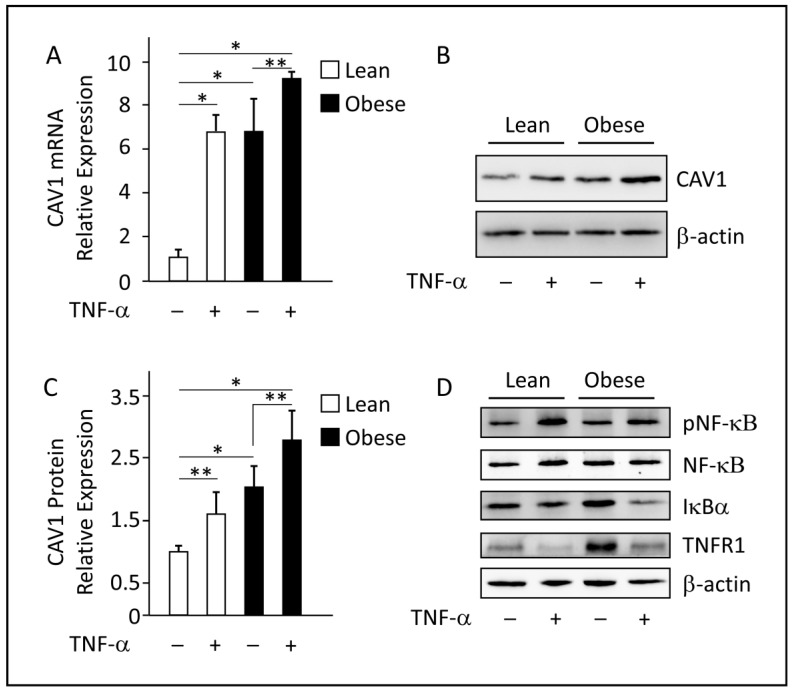
TNF-α augments the expression of CAV1 in differentiated adipocytes isolated from lean and obese individuals. (**A**). Relative to untreated lean adipocytes, *CAV1* mRNA expression levels were significantly elevated in obese adipocytes, and in response to TNF-α treatment for 24 h. Data are presented as mean ± SEM values obtained from three independent experiments (*n* = 3); a Two-tailed unpaired Student’s *t*-test was used to determine significance. (**B**,**C**). Western blots for total protein extracts (5 µg) were developed with antibodies against CAV1 and β-actin, as indicated. CAV1 basal protein expression was significantly elevated in obese compared to lean adipocytes and was significantly induced in response to TNF-α treatments of both lean and obese adipocytes. A two-tailed unpaired Student’s *t*-test was used to determine significance (*n* = 3). * *p* < 0.001; ** *p* < 0.05. (**D**). Treatment with TNF-α activates the NF-κB signaling pathway in both lean and obese adipocytes. Western blots were developed with antibodies against phospho-NF-κB, total NF-κB, IκBα, and the TNF-α receptor TNFR1. 24 h post-TNF-α treatment, NF-κB was activated by phosphorylation and the degradation of its inhibiter IκBα. Furthermore, TNFR1 was desensitized to TNF-α signaling post-24 h treatments (representative Western blots, *n* = 2).

**Figure 5 cells-12-01019-f005:**
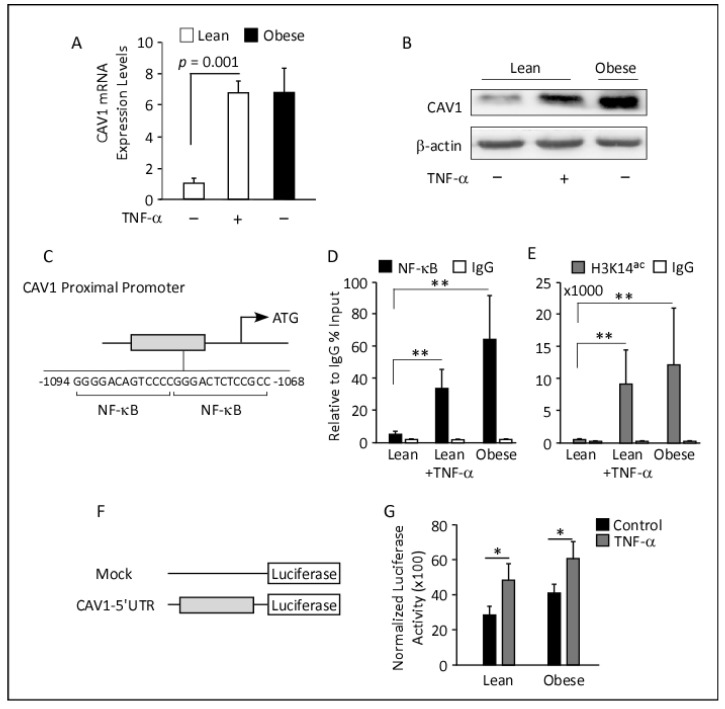
Treatment with TNF-α is associated with an enhanced CAV1 promoter activation. (**A**). *CAV1* transcript levels were significantly elevated in obese pre-adipocytes relative to the untreated lean pre-adipocytes and in response to TNF-α treatment for 24 h. Data from three independent experiments (*n* = 3) are presented as mean ± SEM; significance was determined using a two-tailed unpaired Student’s *t*-test. (**B**). Western blots for total protein extracts (10 µg) were developed with antibodies against CAV1 and β-actin, as indicated. CAV1 basal protein expression was notably elevated in response to TNF-α treatments in lean pre-adipocytes and in obese compared to lean pre-adipocytes. (**C**). Schematic representation of the human *CAV1* locus showing the regulatory region (vertical box) containing NF-κB binding consensus located 1000 bp upstream of the translation start site (arrow). (**D**,**E**). ChIP was performed in pre-adipocytes from lean individuals, either treated with TNF-α or not, and obese individuals, using antibodies against NF-κB or H3K14^ac^ and analyzed by qPCR with primers flanking the NF-κB binding site, as indicated. White bars indicate precipitation with IgG-nonspecific antibodies. Relative enrichment was calculated as the percent chromatin input normalized to IgG from three biological replicas (*n* = 3). ** *p* < 0.01. (**F**). A schematic representation of the *CAV1* gene mapping the proximal regulatory region and showing the cloned 250 bp DNA fragment spanning the NF-κB binding sites into the Luciferase-pRMT-Luc reporter vector. (**G**). Luciferase reporter assays: pre-adipocytes isolated from lean and obese individuals were transfected with a *CAV1*-5′UTR-Luc construct and were either untreated (black bars) or treated with TNF-α. Luciferase reporter activity was notably higher in samples treated with TNF-α. Luciferase activities are presented as a fold change relative to the pRMT-Luc reporter. Data are presented as mean ± SD of three biological replicas (total *n* = 9); a two-tailed unpaired Student’s *t*-test was used to determine significance. * *p* < 0.05.

**Figure 6 cells-12-01019-f006:**
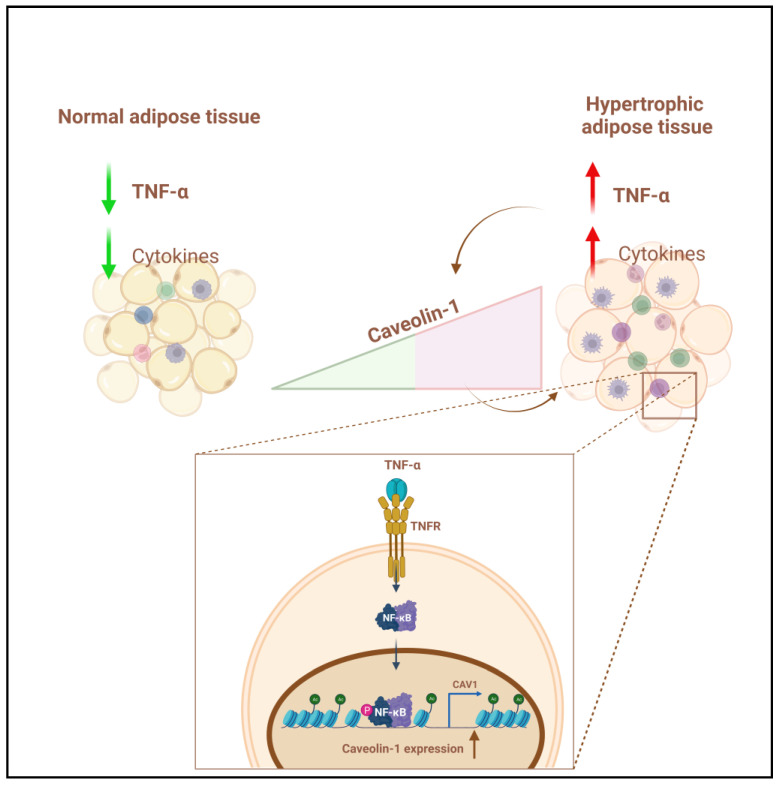
In adipocytes, CAV1 gene expression is regulated by TNF-α/TNFR signaling pathways though the activation of the downstream NF-κB pathway, which in turn binds to its consensus site at the CAV1 regulatory promoter and enhances histone acetylation. Created with BioRender.com.

**Table 1 cells-12-01019-t001:** Anthropometric measurements and clinical characteristics of the study participants.

Phenotype	Lean	Obese	Lean vs. Obese
	(*n* = 7) (Mean ± SD)	(*n* = 20) (Mean ± SD)	(*p*-Value)
Age (years)	43.43 ± 7.45	45.45 ± 12.64	0.695
Weight (kg)	61.37 ± 12.27	93.49 ± 15.45	<0.0001
Height (m)	1.65 ± 0.12	1.65 ± 0.11	0.92
BMI (kg/m^2^)	22.41 ± 2.61	34.40 ± 3.47	<0.0001
Body fat (%)	27.39 ± 6.27	39.31 ± 4.17	<0.0001
Triglycerides (mmol/L)	0.66 ± 0.24	1.32 ± 0.90	0.069
Total cholesterol (mmol/L)	5.40 ± 1.03	5.08 ± 1.27	0.554
HDL cholesterol (mmol/L)	1.67 ± 0.59	1.17 ± 0.29	0.071
LDL (mmol/L)	3.43 ± 0.88	3.35 ± 1.08	0.856
FBG (mmol/L)	5.24 ± 0.53	5.20 ± 0.63	0.86
HbA1C (%)	5.70 ± 0.50	5.59 ± 0.67	0.696
Insulin (mU/L)	6.58 ± 3.33	11.15 ± 7.49	0.211
HOMA-IR	1.51 ± 0.69	2.59 ± 1.86	0.223
WBCs	5.40 ± 1.59	6.235 ± 1.93	0.345

BMI, body mass index; HDL, high-density lipoprotein; LDL, low-density lipoprotein; FBG, fasting blood glucose; HbA1C, glycated hemoglobin C; HOMA-IR, homeostatic model assessment for insulin resistance; WBCs, white blood cells.

**Table 2 cells-12-01019-t002:** Correlation between caveolin-1 gene expression and that of various cytokines or chemokines in non-diabetic obese participants, * *p* < 0.05.

Obese Subjects					
Pearson Correlation (*n* = 20)			Pearson Correlation (*n* = 20)		
	*r*-Value	*p*-Value		*r*-Value	*p*-Value
CC chemokine ligands			CXC chemokine ligands		
CCL2	0.491 *	0.0327	CXCL9	0.544 *	0.0161
CCL3	0.194	0.471	CXCL10	0.500 *	0.0293
CCL8	0.618 *	0.0096	CXCL11	0.552 *	0.0175
CCL15	−0.029	0.905	M1-M2 macrophage transition		
CCL18	0.077	0.748	CD16	0.112	0.647
CCR1	0.574 *	0.016	CD68	−0.004	0.989
CCR2	0.538 *	0.034	CD86	0.128	0.601
Cytokines/interleukins			CD163	0.447 *	0.048
TNF-α	0.547 *	0.0251	Transcription factors related to inflammation		
IL2	0.256	0.277	NF-κB	0.498 *	0.033
IL5	0.053	0.826	Toll-like receptors (TLRs) signaling cascade		
IL6	0.023	0.925	TLR2	0.043	0.879
IL8	−0.069	0.794	TLR3	0.509 *	0.0311
IL10	−0.026	0.915	TLR4	0.488 *	0.058
IL12A	0.015	0.955	TLR7	0.292	0.212
IL13	0.046	0.855	TLR8	0.008	0.975
IL23A	0.083	0.729	TLR9	0.05	0.835
IL1RL1	0.593 *	0.0094	TLR10	0.019	0.937
IL2RA	−0.012	0.96	IRF4	0.263	0.262

## Data Availability

The data are available upon request.

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
