# Peer review of "Adipose Tissue Caveolin-1 Upregulation in Obesity Involves TNF-α/NF-κB Mediated Signaling"

_cells, 2023, doi:10.3390/cells12071019_

Round 1

Reviewer 1 Report

Comments: 

Using subcutaneous AT from 27 (7 lean/20 obese) normoglycemic individuals, in vitro human adipocyte models, and in vivo mice models, the authors found elevated CAV1 expression in obese AT and a positive correlation between the gene expression of CAV1, tumor necrosis factor alpha (TNF-α), and the nuclear factor kappa-light-chain-enhancer of activated B cells (NF-kB). Although the authors presented data relevantly support their conclusion, there are a couple of concerns that needs to be addressed.

1. Authors analyzed CAV1 mRNA level in subcutaneous fat from wildtype and TNF-α KO mice that were intravenously injected with either normal saline or TNF-α. In the method section, authors described that mice were injected intraperitoneally with TNF-α. What is the route for TNFa injection? Authors should add more detailed information for the animal experiment. For example, when to sacrifice the mice after TNF-α injection?

2, Authors analyzed the subcutaneous fat from wildtype and TNF-α KO mice that were intravenously injected with either normal saline or TNF-α. How about other adipose tissue, for example eWAT? How about non-adipose tissue, for example liver? Whether other tissue also have the CAV1 upregulation upon TNFa injection.

3, Author treated preadipocyte from lean and obese individuals with TNFa and found CAV1 mRNA and protein level increased after treatment. Why do authors choose pre-adipcotye to perform this experiment instead of mature adipocyte. In CHIP exp, the author also chooses preadipocyte instead of mature adipocyte. 

4, The authors found the TNFa induced NF-KB signaling and promote CAV1 gene expression. It is strongly suggested to check the NF-KB signaling in TNFa treated mice or cell lines, if possible, to further solid the conclusion of the present study.

5. For figure 1 and 2, authors should add the scale bar to the IHC staining. 

6. Please correct errors and typos. For example, in Table 1, the unit of height should be meter. In line 322 of page 9, “in the subcotaniuos ATs obtained from wildtype” should be “subcutaneous”. In line 305, Figure 2F should be Figure 2E.

7, Genes names are italicized. For example, in line 244 of page 6.

8, in figure 6. Author summarized their finding. In the Figure, Adipokine increased in hypertrophic adipose tissue, which is not accurate. Adipokine is secreted by adipose tissue. There are hundreds of adipokine in adipose tissue. Not all of them increased in obesity. For example, adiponectin level decreased in people with obesity. 

Author Response

Comments and Suggestions for Authors

Comments: 

Using subcutaneous AT from 27 (7 lean/20 obese) normoglycemic individuals, in vitro human adipocyte models, and in vivo mice models, the authors found elevated CAV1 expression in obese AT and a positive correlation between the gene expression of CAV1, tumor necrosis factor alpha (TNF-α), and the nuclear factor kappa-light-chain-enhancer of activated B cells (NF-kB). Although the authors presented data relevantly support their conclusion, there are a couple of concerns that needs to be addressed.

  1. Authors analyzed CAV1 mRNA level in subcutaneous fat from wildtype and TNF-α KO mice that were intravenously injected with either normal saline or TNF-α. In the method section, authors described that mice were injected intraperitoneally with TNF-α. What is the route for TNFa injection? Authors should add more detailed information for the animal experiment. For example, when to sacrifice the mice after TNF-α injection?.

We thank the reviewer for this comment. We have made a mistake and corrected it in the revised manuscript. To clarify, mice were given intraperitoneal injections of saline or TNF-a 50 ug/kg for 5 days, followed by 100 ug/kg for 3 days. The dose injections were designed based on preliminary results for the induction of CAV1 and other factors (please see Pages 5–6, lines 224-228).

 2, Authors analyzed the subcutaneous fat from wildtype and TNF-α KO mice that were intravenously injected with either normal saline or TNF-α. How about other adipose tissue, for example eWAT? How about non-adipose tissue, for example liver? Whether other tissue also have the CAV1 upregulation upon TNFa injection. 

We would like to thank the reviewer for the requested clarification.

In this study, the only available human tissue samples were from subcutaneous adipose tissue and human subcutaneous pre-adipocytes; therefore, we focused on this tissue type in the animal experiments and throughout the manuscript. Furthermore, we focused on the adipocytes because they are the major cell type in adipose tissue and are implicated in metabolic disorders.

3, Author treated preadipocyte from lean and obese individuals with TNFa and found CAV1 mRNA and protein level increased after treatment. Why do authors choose pre-adipcotye to perform this experiment instead of mature adipocyte. In CHIP exp, the author also chooses preadipocyte instead of mature adipocyte.

We would like to thank the reviewer for the requested clarification. Our initial studies showed that TNF-a treatments induce CAV1 expression in both pre-adipocytes and adipocytes. In the CHIP analyses, we used pre-adipocytes because these studies require a lot of cells, and many T175 flask to collect chromatins and immuno-precipitations using NF-kb, Histone H3K14ac or IgG control antibodies. Similarly, luciferase assay, include transfections and several repeated experiments to get statistical data. In the experiments for gene expression at RNA and protein levels we used differentiation mature adipocytes.   

4, The authors found the TNFa induced NF-KB signaling and promote CAV1 gene expression. It is strongly suggested to check the NF-KB signaling in TNFa treated mice or cell lines, if possible, to further solid the conclusion of the present study.

We would like to thank the reviewer for the requested clarification. As suggested, we studied the TNFa/NF-kb pathway proteins, including NF-kb, phosphor-NF-kb, IkBa, and TNFR1. Please see the results section (Page 10, lines 356-364), the introduced Figure 4D (Page 11, lines 373-376), and the discussion section (Page 13, lines 422-433) 

  1. For figure 1 and 2, authors should add the scale bar to the IHC staining. 

We would like to thank the reviewer for his or her input. The scale bars were small and unclear. We modified Figures 1 and 2 and clarified the scale bars.

  1. Please correct errors and typos. For example, in Table 1, the unit of height should be meter. In line 322 of page 9, “in the subcotaniuos ATs obtained from wildtype” should be “subcutaneous”. In line 305, Figure 2F should be Figure 2E.

We would like to thank the reviewer for the input. We agree, and the table is corrected accordingly. The typo error is corrected in line 325 of page 9 and in line 311 of Figure 2E.

7, Genes names are italicized. For example, in line 244 of page 6.

We would like to thank the reviewer for the feedback. We corrected all the gene names and italicized them accordingly throughout the manuscript (red font).

8, in figure 6. Author summarized their finding. In the Figure, Adipokine increased in hypertrophic adipose tissue, which is not accurate. Adipokine is secreted by adipose tissue. There are hundreds of adipokine in adipose tissue. Not all of them increased in obesity. For example, adiponectin level decreased in people with obesity. 

We would like to thank the reviewer for his/her input. We agree, the figure is corrected accordingly.

Reviewer 2 Report

Major concerns (comments on scientific content)

1. For Fig. 1, the images inset and the actual image seems to be the same magnification by looking at the overall sizes of adipocytes. Could the authors please confirm if the magnifications mentioned here are the correct ones? Also, could the authors please point out precisely which area is magnified here? Although it is easy to notice by looking at the overall morphology of the adipocytes within the field, it would make reading the image easier. Also, the color resolution between lean and obese staining images seems different. Is it because of the different levels and resultant staining intensity of CAV1?

2. The above comments are also applicable to Fig. 2. Also, could the authors please explain whether the CAV1 staining was performed again on the same samples as the ones showing TNF-α staining to produce the graph in panel E? In other words, could the authors also include the images showing CAV1 staining that were used to produce the graph in panel E?

3. In results section 3.5, line 320, the authors mention that TNF-α has been shown to be the main regulator of CAV1 expression from the data obtained so far. However, the data obtained so far only shows the correlation between TNF-α, NF-kB and CAV1, and not the causation. It does not show that the TNF-α is the regulator of CAV1 expression. The authors will need to re-phrase this statement. Also, in lines 323-324, the authors mention to have reported elevated CAV1 mRNA and protein in TNF-α KO as well as WT mice. However, from the graphs, the mRNA seems to be non-significant whereas the protein level seems unchanged. Could the authors please explain this?

4. In results section 3.5, line 322, please change the word “subcotaniuos” to “subcutaneous”.

5. For Fig. 3, have the authors determined the levels of CAV1 in other adipose tissue depots such as eWAT which is an example of visceral AT? Is there a rationale of including or studying only the subcutaneous AT? Additionally, could the authors provide an explanation as to why the CAV1 protein did not go up in WT mice with TNF-α treatment?

6. In results section 3.7, line 366 and related description, was this analysis also performed on differentiated adipocytes? Could the authors include that data as well? If it was indeed only pre-adipocytes, could the authors please provide a rationale for it? Do the authors feel the CAV1 expression going up is from the pre-adipocytes both in-vivo and in-vitro? In section 3.6, the authors mention to have performed experiments on “differentiated pre-adipocytes”, whereas in section 3.7, they mention only pre-adipocytes. If the authors indeed differentiated the pre-adipocytes obtained from patients, please include a differentiation protocol in methods. Also, the figure suggests that the pre-adipocytes (or adipocytes, since it is not clear) from obese individuals were not given TNF-alpha treatment. Whereas the statement in line 366-367 seems to indicate that even the pre-adipocytes (or adipocytes) from obese individuals were treated with TNF-alpha. Could the authors please clarify this?

Author Response

Reviewer 2

Comments and Suggestions for Authors

Major concerns (comments on scientific content)

  1. For Fig. 1, the images inset and the actual image seems to be the same magnification by looking at the overall sizes of adipocytes. Could the authors please confirm if the magnifications mentioned here are the correct ones? Also, could the authors please point out precisely which area is magnified here? Although it is easy to notice by looking at the overall morphology of the adipocytes within the field, it would make reading the image easier. Also, the color resolution between lean and obese staining images seems different. Is it because of the different levels and resultant staining intensity of CAV1?

We would like to thank the reviewer for the requested clarification. The magnifications are correct: 20X and 40X (insert), but due to resizing the image to fit, the insert image lost its integrity. To avoid any confusion, we deleted the insert. Yes, the differences are due to the lower CAV1 expression in lean. Since these images are representative of other 5 images. We decided to choose another image in which the background is comparable between tissues from lean and obese individuals.

  1. The above comments are also applicable to Fig. 2. Also, could the authors please explain whether the CAV1 staining was performed again on the same samples as the ones showing TNF-α staining to produce the graph in panel E? In other words, could the authors also include the images showing CAV1 staining that were used to produce the graph in panel E?

We would like to thank the reviewer for the requested clarification. As described in comment 1. To avoid any confusion, we deleted the insert. Yes, CAV1 and TNF-a IHCs were done using the same samples. We mentioned this information in the Figure 2 legend (now highlighted in yellow, page 9, lines 319-320). We did not include CAV1 images in Figure 2 to avoid image duplication in both Figures 1 and 2.

  1. In results section 3.5, line 320, the authors mention that TNF-α has been shown to be the main regulator of CAV1 expression from the data obtained so far. However, the data obtained so far only shows the correlation between TNF-α, NF-kB and CAV1, and not the causation. It does not show that the TNF-α is the regulator of CAV1 expression. The authors will need to re-phrase this statement. Also, in lines 323-324, the authors mention to have reported elevated CAV1 mRNA and protein in TNF-α KO as well as WT mice. However, from the graphs, the mRNA seems to be non-significant whereas the protein level seems unchanged. Could the authors please explain this?

We would like to thank the reviewer for the feedback. We agree that until section 3.5, we showed correlations between the factors. Our conclusion was based on the multiple stepwise regression where the association between CAV1 and TNF-a was significant (β = 0.06; p = 0.032; Page 8, lines 302-304, highlighted in Yellow). Nevertheless, we re-phrased the sentence to avoid confusion to the readers.

Yes, we agree with the Reviewer. CAV1 expression was significantly elevated in the TNF-a KO mice post TNF-a administration. We corrected the sentence (Please see Page 9, Lines 328-335).   

  1. In results section 3.5, line 322, please change the word “subcotaniuos” to “subcutaneous”.

We would like to thank the reviewer for the input. We agree, and the table is corrected accordingly. The typo error is corrected (line 325 of page 9).

  1. For Fig. 3, have the authors determined the levels of CAV1 in other adipose tissue depots such as eWAT which is an example of visceral AT? Is there a rationale of including or studying only the subcutaneous AT? Additionally, could the authors provide an explanation as to why the CAV1 protein did not go up in WT mice with TNF-α treatment?

We would like to thank the reviewer for the requested clarification. We did not study CAV1 expression in different tissues. This is a great idea and will be done in the near future.

In this study, the only available human tissue samples were from subcutaneous adipose tissue and human subcutaneous pre-adipocytes; therefore, we focused on this tissue type in the animal experiments and throughout the manuscript. Furthermore, we focused on the adipocytes because they are the major cell type in adipose tissue and are implicated in metabolic disorders.

The reason why CAV1 did not go up significantly in wild-type (WT) mice after treatment with exogenous TNF-a is not entirely clear until now. It is possible that the levels of endogenous TNF-a in the WT mice were already sufficient to maintain high levels of CAV1 expression, so the additional administration of TNF-a did not have a significant effect on CAV1 expression. Another possibility is that the effect of TNF-a on CAV1 expression may be more pronounced in the absence of other downstream factors that could modulate CAV1 expression, such as other cytokines or growth factors affected by TNF-a pathway.

  1. In results section 3.7, line 366 and related description, was this analysis also performed on differentiated adipocytes? Could the authors include that data as well? If it was indeed only pre-adipocytes, could the authors please provide a rationale for it?

Our initial studies showed that TNF-a treatments induced CAV1 expression in both pre-adipocytes and adipocytes. In the CHIP analyses, we used pre-adipocytes because these studies require a lot of cells, and many T175 flask to collect chromatins and immuno-precipitations using NF-kB, Histone H3K14ac or IgG control antibodies. Similarly, luciferase assays include transfections that cannot be done on differentiated adipocytes. In the experiments for gene expression at RNA and protein levels, we used differentiated adipocytes.

Do the authors feel the CAV1 expression going up is from the pre-adipocytes both in-vivo and in-vitro?

Yes, CAV1 expression is indeed increased in pre-adipocytes and adipocytes in both in vivo and in vitro in response to TNF-a,

Elevated levels of CAV1 expression, in response to TNF-a, is in both pre-adipocytes and adipocytes based on our preliminary data (not shown). Adipocytes are the main component of adipose tissue, as shown in the manuscript Figures CAV1 is responsive to TNF-a administration in mature adipocytes.

In section 3.6, the authors mention to have performed experiments on “differentiated pre-adipocytes”, whereas in section 3.7, they mention only pre-adipocytes. If the authors indeed differentiated the pre-adipocytes obtained from patients, please include a differentiation protocol in methods.

The differentiation protocol is included in the material and Methods sections. Briefly, we incubated the primary cells in the differentiation and maintenance media supplied by ZEN-Bio, in accordance with their guidelines (Page 5, lines 199-201).

Also, the figure suggests that the pre-adipocytes (or adipocytes, since it is not clear) from obese individuals were not given TNF-alpha treatment. Whereas the statement in line 366-367 seems to indicate that even the pre-adipocytes (or adipocytes) from obese individuals were treated with TNF-alpha. Could the authors please clarify this?

Please be advised that the ChIP analysis were done on pre-adipocytes. TNF-a treatment was done for the lean pre-adipocytes only, and not the obese pre-adipocytes. We re-phrased the whole section 3.7 to clarify the experiments and the objecti

Round 2

Reviewer 1 Report

Authors addressed all my concerns. 

Author Response

We would like to thank the reviewer for his/her help in improving our manuscript.

Reviewer 2 Report

1. I have a concern regarding performing ChIP experiments in pre-adipocytes instead of adipocytes. The authors have indeed seem to have successfully differentiated lean and obese pre-adipocytes into adipocytes to show western blot data. They have also included differentiation protocol in Methods. Could the authors show some images as to what percentage of pre-adipocytes turned into adipocytes? OR could the authors include data for adipogenic markers such as adiponectin, leptin, perilipin? If the authors were successful in generating good differentiation, then they can also perform ChIP experiments on differentiated adipocytes. Also, since the authors were able to get significant number of pre-adipocytes for ChIP experiments, expansion of pre-adipocytes does not seem to be a problem. And if getting large number of pre-adipocytes is possible, then getting good number (not 100%) of differentiated adipocytes should also be possible. 

If the authors have found the CAV1 expression to be elevated even in pre-adipocytes, could they include it in the manuscript or briefly mention it to justify using pre-adipocytes for ChIP experiments?

2. Could the authors provide description as to how long the adipocytes were differentiated?

Author Response

Comments and Suggestions for Authors

  1. I have a concern regarding performing ChIP experiments in pre-adipocytes instead of adipocytes. The authors have indeed seem to have successfully differentiated lean and obese pre-adipocytes into adipocytes to show western blot data. They have also included differentiation protocol in Methods.

A. Could the authors show some images as to what percentage of pre-adipocytes turned into adipocytes? OR could the authors include data for adipogenic markers such as adiponectin, leptin, perilipin? If the authors were successful in generating good differentiation, then they can also perform ChIP experiments on differentiated adipocytes.

We would like to thank the reviewer for his/her help in improving our manuscript. The differentiation efficiency is 50-60%, mature adipocytes. We detailed the differentiation protocol in the Materials and Methods section and included a figure for the requested Western blots (adiponectin and perilipin1) and images pertaining to the differentiated cells as supplementary data. (Please see Page 5, lines 200-208, and Supplementary Figure S2) Please note that the Western blots protein loadings and exposure time are different than that in Figure 4. Further, CAV1 protein levels were also induced in response to adipogenesis (Figure S2).

B. Also, since the authors were able to get significant number of pre-adipocytes for ChIP experiments, expansion of pre-adipocytes does not seem to be a problem. And if getting large number of pre-adipocytes is possible, then getting good number (not 100%) of differentiated adipocytes should also be possible.

It is important to note that the responses of pre-adipocytes and adipocytes to TNFa treatments are comparable, and we used different experimental approaches to show the impact of TNFa in these cells.

  • For example, the Luciferase assays that were done using pre-adipocytes clearly show the reporter activity post-treatment with TNFa at the CAV1 locus harboring the NF-kB binding site.

  • The differentiation efficiency is 50-60%, mature adipocytes, as indicated by the accumulation of lipid droplets, suggesting that the culture is a mix of pre-adipocytes, adipocytes, and intermediate cells (adipocytes with small and no visible accumulated lipid droplets). Together, this cell population is responsive to TNFa treatment with notable CAV1 expression enhancement.

As we previously explained, a ChIP experiment using at least three antibodies (NF-kB, acetylated-H3, and control IgG) requires a minimum of 200ug purified chromatin from 3 x 150 cm tissue culture plates. Our experiments are done using two human primary cell lines, with and without treatment, and in triplicate for proper statistical significance analysis. Please keep in mind that culturing primary cell lines is a slow process. In addition, the differentiation protocol requires at least another 12–14 days of daily monitoring. All these factors require a lot of material, effort, and time, and the conclusion will be the same: The TNFa/NF-kB pathway induces CAV1 expression.

C. If the authors have found the CAV1 expression to be elevated even in pre-adipocytes, could they include it in the manuscript or briefly mention it to justify using pre-adipocytes for ChIP experiments?

We would like to thank the reviewer for his/he insight. We included the requested Western blots and RT-PCR data studies previously done on pre-adipocytes (Please see page 11, lines 385-386 and Figures 5A and B).

  1. Could the authors provide description as to how long the adipocytes were differentiated?

We thank the reviewer for the requested clarification. The differentiation protocol takes about 12 to 14 days. We detailed the protocol in the Materials and Methods section to avoid any miss interpretation (Please see Page 5, lines 200-208, and Supplementary Figure S2).